# Investigating Effects of Landfill Soil Gases on Landfill Elevated Subsurface Temperature

**Samain Sabrin [1]**, **Rouzbeh Nazari [1,2]**, **Md Golam Rabbani Fahad [1]**, **Maryam Karimi [1,2]**, **Jess W. Everett [3]** and **Robert W. Peters [1,\*]**

1   Department of Civil Construction and Environmental Engineering, The University of Alabama at Birmingham, Hoehn Engineering Building, 1075 13th Street South, Birmingham, AL 35294-4440, USA; sabrins8@uab.edu (S.S.); rnazari@uab.edu (R.N.); fahadr@uab.edu (M.G.R.F.); karimi@uab.edu (M.K.)

2   Department of Environmental Health Science, The University of Alabama at Birmingham, School of Public Health, Ryals Public Health Building (RPHB), 1665 University Boulevard, Birmingham, AL 35294-0022, USA

3   Department of Civil & Environmental Engineering, Henry M. Rowan College of Engineering, Rowan University, Glassboro, NJ 08071, USA; everett@rowan.edu

*   Correspondence: rwpeters2004@gmail.com



**Featured Application: This research was designed to understand the effects of landfill soil gases on the events of subsurface landfill fires or elevated temperature which is experienced by many landfills during their operational and post-operational period. A framework proposed in this article can assist landfill authorities in controlling or preventing such hazardous incidents related to subsurface elevated thermal conditions.**

**Abstract:** Subsurface temperature is a critical indicator for the identification of the risk associated with subsurface fire hazards in landfills. Most operational landfills in the United States (US) have experienced exothermic reactions in their subsurface. The subsurface landfill area is composed of various gases generated from chemical reactions inside the landfills. Federal laws in the US mandate the monitoring of gases in landfills to prevent hazardous events such as landfill fire breakouts. There are insufficient investigations conducted to identify the causes of landfill fire hazards. The objective of this research is to develop a methodological approach to this issue. In this study, the relationship was investigated between the subsurface elevated temperature (SET) and soil gases (i.e., methane, carbon dioxide, carbon monoxide, nitrogen, and oxygen) with the greatest influence in landfills. The significance level of the effect of soil gases on the SET was assessed using a decision tree approach. A naïve Bayes technique for conditional probability was implemented to investigate how different gas combinations can affect different temperature ranges with respect to the safe and unsafe states of these gases. The results indicate that methane and carbon dioxide gases are strongly associated with SETs. Among sixteen possible gas combinations, three were identified as the most probable predictors of SETs. A three-step risk assessment framework is proposed to identify the risk of landfill fire incidents. The key findings of this research could be beneficial to landfill authorities and better ensure the safety of the community health and environment.

**Keywords:** elevated temperature; emissions; gases; landfill fire; methane; risk

---

## 1. Introduction

To date, landfills have been the most commonly used solid-waste disposal option. Solid-waste landfills have been designed to contain municipal solid waste in a large scale containment system to prevent and control the harmful effects on the environment and surrounding communities. A total

of 262 billion kilograms of municipal solid waste were assessed to be produced each year in the United States (US), and approximately 52.7% of this waste is buried within landfills [1]. In recent decades, the rate of municipal solid waste (MSW) production has increased substantially in the US, from 211.64 billion kilograms in 1990 to 254.42 billion kilograms in 2011 [2], whereas the number of landfills has substantially declined, from about 8000 in 1988 [3] to 1908 in 2011 [2]. This decline in landfill numbers is mostly because of the strict guidelines enforced by the Environmental Protection Agency (EPA), which has led to accommodating the increasing amount of MSW along with large waste piles within the few remaining landfills and resulting in the smaller surface-to-volume ratios. When the self-heat produced in the natural biodegradation processes exceeds the rate of heat dissipation through the landfill surface, temperatures may increase such that the landfill can ignite spontaneously [4,5].

This article mainly focuses on subsurface fire events of the MSW landfills. Landfills are prone to subsurface fires because of their unique compositions and construction. The National Fire Incident Reporting System (NFIRS), reported in [6], found that more than 25% of these incidents occurred repeatedly at the same landfills [7]. Landfill fires can be classified as surface and subsurface fires. Surface fires that ignite over noncompacted freshly covered solid waste can occur due to different reasons such as discarding of the smoldering materials, spontaneous ignition and improper control systems of landfill-gases. Any combustion happening under the ground level within the waste mass is defined as a subsurface combustion which is not visible at the ground surface. These fires may continue undetected for many years; hence it becomes difficult to determine the extent of landfill damage. These events can cause tremendous internal and structural damage by consuming large amounts of waste and may result in the collapse of landfill sections while personnel are actively engaged in an attempt to contain the fire [8]. A smoldering event is initiated when temperatures are high enough, which can emit air pollutants including but not limited to volatile organic compounds (VOCs), semivolatile organic compounds (SVOCs), carbon monoxide (CO), polycyclic aromatic hydrocarbons (PAHs), and pose serious environmental and health hazards [9–13]. Subsurface combustion in landfills may cause severe damage to the leachate collection and liner system. Most subsurface fires burn slowly without showing visible flame, making it more difficult to detect compared to the surface landfill fires. Detecting subsurface fires is generally very difficult unless smoke or areas of settlement are observed, or a continuous measurement system is in place that records gas levels. A recent and existing example of such events is a dumpsite underground fire in the Forestdale area of Alabama, which has continued to burn since May 2020, damaging the ground stability and covering this residential area with hazardous smoke [14]. In such cases, the levels of CO (1000 ppm or more) are higher, well temperatures exceed 170 °F (76.67 °C) or the gas temperature within the extraction system is over 140 °F (60 °C).

Elevated temperatures in landfills can occur by events characterized by high temperatures near the surface or in the subsurface at depths of greater than 20 m [15,16]. Such events have been observed in all types of landfills and can affect the gas composition, leachate quality, slope stability, and the liner system and integrity of the landfill, also posing substantial environmental hazards by their release of by-products from incomplete combustions [10–12,17–28]. Some case studies indicate the progression of subsurface elevated temperature (SET) with elevated waste, gas wellhead temperatures and reduced methane-to-carbon dioxide ratios with a consequent rise in the generation and accumulation of hydrogen gases and CO [16,29]. Rees [30] considered the control of temperature by optimizing methane production and waste decomposition. Yeşiller et al. [31] investigated the thermal conditions in MSW landfills and described these characteristics as a function of landfill operational conditions and climatic regions. The periodic gas production and temperature change in MSW landfills have been inspected and the slope stability was analyzed based on increased gas–liquid pressures along with elevated temperatures [32,33]. Jafari et al. [34] presented a case study of the changes in gas compositions, slope movement, settlement, and leachate migration and developed indicators associated with the progression of SETs. However, the existing literature does not address how subsurface temperature behaves in the presence of several risk parameters and gradually elevates from one combination of gas parameters to another. This article asks questions on how a given set of gas combinations yields

temperatures in the subsurface environment besides observing the effects of individual gas parameters. This paper focuses on a methodical evaluation of risk factors affecting the corresponding SET that indicates possible fires. The aim of this paper is to discuss the unsafe ranges of subsurface temperatures with potential to cause substantial damage to the landfill consistency based on safe and unsafe soil gas ranges in the landfill gas collection system.

## 2. Background

Volumetrically, landfill soil gases usually comprises 40 to 60% carbon dioxide ($CO_2$), 45 to 60% methane ($CH_4$), 2 to 5% nitrogen ($N_2$), and 0.1 to 1% oxygen ($O_2$) along with low fractions of ammonia, CO, hydrogen, sulfides, and nonmethane organic compounds (NMOCs) such as benzene, trichloroethylene, and vinyl chloride [35]. Landfills produce gases in three processes: most gases are produced by aerobic and anaerobic bacterial decomposition; solid or liquid wastes convert into vapor via volatilization, e.g., nonmethane organic compounds (NMOCs) from disposal chemicals; chemical reactions between the remaining chemicals in waste [35]. Over a period of decades, landfill waste goes through bacterial decomposition in five phases:

Phase I: Refuse matters are converted into water and $CO_2$ in the aerobic decomposition process by aerobic bacteria that live in an oxygen enriched environment.

Phase II: In the absence of oxygen, anaerobic bacteria initiate anaerobic decomposition.

Phase III: Some anaerobic bacteria produced in Phase II consume the organic acids in this decomposition phase. Organic acids formed via a microbial hydrolysis process convert to soluble acids and further break down to nitrous oxide, $CO_2$, $CH_4$ and a small amount of VOCs [36,37].

Phase IV: This phase is relatively steady with the production rates and composition of the landfill gases. These gases are produced at a steady rate for 50 or more years after the waste pile is placed in the landfill [38].

Phase V: Some organic matters are available to be decomposed and oxygen is introduced in the environment.

Figure 1 shows a flow chart of the chemical decomposition processes and the by-products generated during the different phases of waste decomposition. Factors such as waste characteristics (age and composition of the refuse), disposed chemicals in landfills, and some environmental factors (moisture content, temperature, the presence of oxygen) influence the volume and rate of generated landfill gases (e.g., $CO_2$, $CH_4$, $N_2$, and hydrogen sulfide) [39]. Bacterial activity increases as the landfill's temperature rises resulting in increased gas production. Changes in gas compositions have been observed to occur in advance of increases in wellhead temperatures, which can be an indicator of an approaching SET event [34].

MSW landfills produce $CO_2$, water, and heat in the process of aerobic decomposition (Jafari et al., 2017a). In the absence of oxygen, anaerobic decomposition initiates after aerobic decomposition with the resultant production of $CO_2$, $CH_4$, and heat. Equations (1) and (2) can be used to express the aerobic and anaerobic transformation of organic waste, respectively [34]. A comparison between the enthalpies of both reactions indicates that the heat generated in aerobic decomposition is approximately 19 times higher than the heat produced from an anaerobic reaction [34]. As a result, the temperature typically ranges between 60 and 80 °C for a waste pile in aerobic conditions [34,40], whereas the temperature in the anaerobic landfills ranges from approximately 25 to 45 °C [31,33]. Heat accumulation by some exothermic reactions with the intrusion of oxygen or aerobic degradation creates a suitable condition to initiate and continue the smoldering combustion of MSWs [20,39]. The tetrahedron of combustion theory describes the four conditions required for a combustion to happen [39,41]: (1) an oxidizing agent (e.g., oxygen via air intrusion); (2) a source of fuel in MSWs; (3) a source of energy (e.g., heat generated from any exothermic reaction or aerobic decomposition); (4) chain reaction of combustion that is self-sustaining (e.g., charred waste). In MSW landfills, it is essential to limit any air intrusion that can be readily controlled. Subsurface combustion in landfills usually spreads through smoldering

combustion directly occurring on the solid fuel surface [10]. CO, $CO_2$, water vapor, and heat are yielded from the partial smoldering combustion of cellulose [42], as shown in Equation (3).

$$C_6H_{12}O_6 + 6O_2 \rightarrow 6CO_2 + 6H_2O \; ; \; \Delta H = -2815 \text{ kJ/mol} \tag{1}$$

$$C_6H_{12}O_6 \rightarrow 3CO_2 + 3CH_4 \; ; \; \Delta H = -145 \text{ kJ/mol} \tag{2}$$

$$C_6H_{10}O_5 \text{ (s)} + 5.7O_2 \text{ (g)} \rightarrow 5.4CO_2 \text{ (g)} + 0.6CO \text{ (g)} + 5H_2O \; ; \; \Delta H = -2440 \text{ kJ/mol} \tag{3}$$

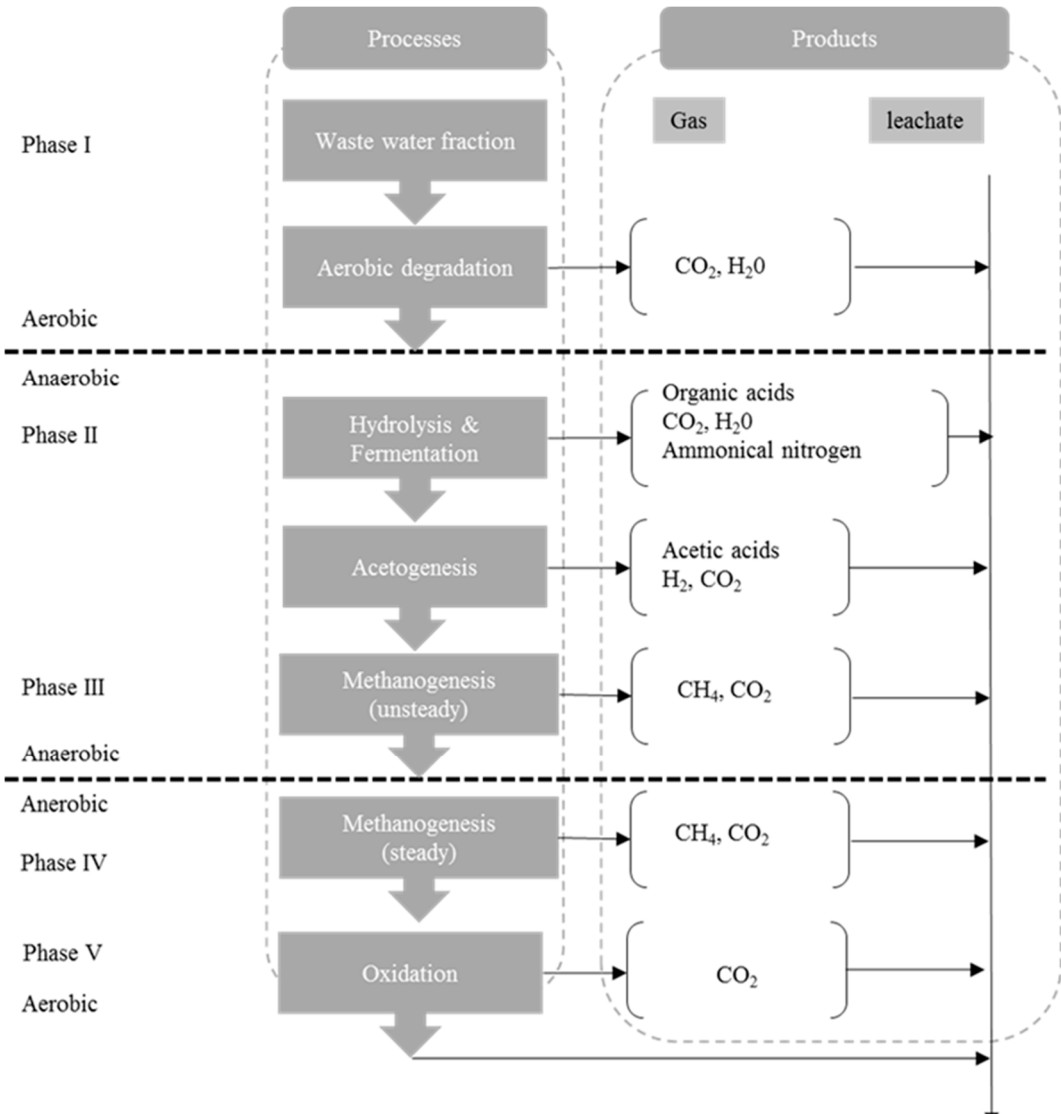

**Figure 1.** Chemical processes and byproducts generated in 5 phases of bacterial decomposition.

Smoldering ignition does not proceed to completion due to the limited $O_2$ available but can propagate at a low level of oxygen, e.g., less than 3% volume-to-volume ratio (*v/v*) [43–45]. Smoldering combustion within an MSW landfill has been documented to persist at temperatures between 100 and 120 °C [46]. In some cases, the temperature during smoldering combustion has been detected in the range of 200–300 °C and even as high as 700 °C in MSW landfills [23,28]. Bergström and Björner [47] measured a deep subsurface fire with a range of 80–230 °C. Research has shown that the integrity and service life of landfill leachate control systems, gas-extraction systems, materials and cover of the composite liner systems can be impacted by a sustained temperature as low as 85 °C [48]. During a

SET, the landfill gas composition rapidly changes from $CO_2$ (40–55% *v/v*) and $CH_4$ (50–60% *v/v*) to a gas composition of $CO_2$ (60–80% *v/v*), hydrogen (10–35% *v/v*), CO (>1500 ppmv) along with the ratio of $CH_4$ to $CO_2$ falling below 1 [34]. Jafari et al. [49] observed the subsurface temperature rising from 55 to 75 °C in their case study, when $CH_4$ to $CO_2$ ratio reduced from 1.2 to 0.3 in a course of 50 days [49].

The technical procedures for detecting, evaluating, and mitigating elevated temperatures or landfill fires vary in the literature. The elevated temperature considered in this study refers to an increase in gas-well temperature beyond a certain threshold. The United States Environmental Protection Agency (US EPA) recommends a normal level of $O_2$, $CH_4$, and temperature ranges for landfills to be, respectively, <5%, 45% to 60%, and less than <55 °C, and a CO level of 2000 ppm is regarded as an action level (the concentration level in excess requires regulatory or remedial action) [50]. The Solid Waste Association of North America (SWANA) suggests temperature <52 °C, $O_2$ levels <1%, and $CH_4$ levels from 45 to 58% as normal ranges, and traces of CO in excess of 25 ppm calls for an action level to take precautionary steps against SETs [39,51]. The difference in the ranges for $O_2$, $CH_4$, and temperature recommended by the SWANA from those of the US EPA are minimal. The level for CO suggested by the US EPA is more tolerant, up to 2000 ppm, whereas the SWANA has a strict interpretation of the action level with the presence of only 25 ppm CO. Unlike the US EPA, the SWANA considers CO and residual nitrogen ($RN_2$) to be possible indicators of smoldering events. Table 1 displays the ranges of subsurface $RN_2$ level along with their indications as described by Estrabrooks [52]. The SWANA emphasizes the monitoring of CO to assess landfill fires. Table 2 simplifies the details regarding landfill operations and fire prevention. Other organizations such as the Ohio Environmental Protection Agency (Ohio EPA), International Solid Waste Association (ISWA), U.S. Army Corps of Engineers (USACE), Republic Services, Inc., Cornerstone, Geosyntec Consultants, Conestoga-Rovers and Associates provide guidance for landfill gas management practices and the management of smoldering events in the form of standard operating procedures. Some of the criteria in their documents overlap and others vary from organization to organization. However, these recommendations can be only used in the presence of a fire incident already endangering the whole landfill system and surrounding environment, which implies incompetency in the current system. A risk assessment method is crucial in associating these parameters to the future risks of SETs so that fire outbreaks can be predicted and prevented.

**Table 1.** Ranges of residual nitrogen ($RN_2$) in the landfill soil gases composition [52].

| Percentage of Residual Nitrogen ($RN_2$) | Indications |
| --- | --- |
| 0–12% | The internal extraction system contains this range in most operating landfills. |
| 16–20% | Deemed essential to regulate perimeter migration, side slope emission, or where other compromises are required. |
| >20% | Indication of an aggressive landfill gas-extraction system with potential to initiate aerobic conditions. |

**Table 2.** Recommendation by the Solid Waste Association of North America (SWANA) and the United States Environmental Protection Agency (US EPA) regarding landfill management and fire prevention in the U.S. (Thalhamer, 2013).

| Document | Recommended /Allowed Oxygen Intrusion (%) | Normal Methane Range (%) | Temperature Action Range (°C) | Carbon Monoxide (CO) Action Level (ppm) | Symptoms/Indications of a Smoldering Event or Comments |
|---|---|---|---|---|---|
| Solid Waste Association of North America (SWANA, 1997) | 0 to 0.5% (Ideal range)  <1% (Maximum limit) | 45 to 58% | 15.5 °C to 51.7 °C  (Typical range)  51.7 °C to 60 °C (Action level) | Trace  <25 ppm | • CO is an indicator of the possible presence of a subsurface fire<br>• a byproduct of incomplete combustion<br>• indicator of a possible SET or fire.<br>• Can be a testing parameter for landfill fire<br>• Temperature limit for PVC is 165 °F<br>• Starving the fir of oxygen is the best way to handle a landfill gas fire<br>• High residual $N_2$ levels may be an indication of a landfill fire<br>• Landfill gases can be in the combustible range within the gas collection piping if oxygen is sufficiently high around 10% or greater. |
| United States Environm-ental Protection Agency (US EPA)  (Robertson and Dunbar, 2005) | 0.1 to 1% (Ideal range)  <5% (Maximum limit) | 45 to 60% | >55 °C (Action level) | 0 to 2000 ppm | • Excessive air intrusion into the landfill wastes can initiate SET and subsurface fire.<br>• There must be data demonstrating that the elevated parameter(s) does not cause fires or significantly inhibit anaerobic decomposition of the waste (40 CFR §60.753, 2002) |

## 3. Methodology

This research was conducted in the following steps that include the data collection of landfill gases (such as $CH_4$, $CO_2$, $O_2$ and balance gas), the categorization of parameters (gases and wellhead temperature) in terms of their safe ranges, and the performance of statistical tests to assess each parameter's influence on temperature. The statistical tests included the conditional inference trees algorithm to assess the effects of all parameters in different temperature ranges [53]. The significance levels of the soil gas parameters affecting the SET were assessed using this decision tree approach, which can be used to detect the most significant parameters affecting landfill subsurface fires. Next, the influence of various gas parameter combinations on the subsurface temperatures was analyzed. A naïve Bayes technique [54] was utilized to determine how different gas combinations can affect different temperature ranges in terms of safe and unsafe gas states. The likelihood of different temperature ranges corresponding with the possible parameter combinations was investigated to identify the combinations primarily responsible for SETs. Both the conditional inference trees algorithm and the naïve Bayes conditional probability tests were performed using "R", a programming language for statistical computing. Lastly, a three-step risk assessment framework is proposed to identify the risk of landfill fire incidents. Figure 2 shows a flow chart of the step-by-step research procedure and outputs.

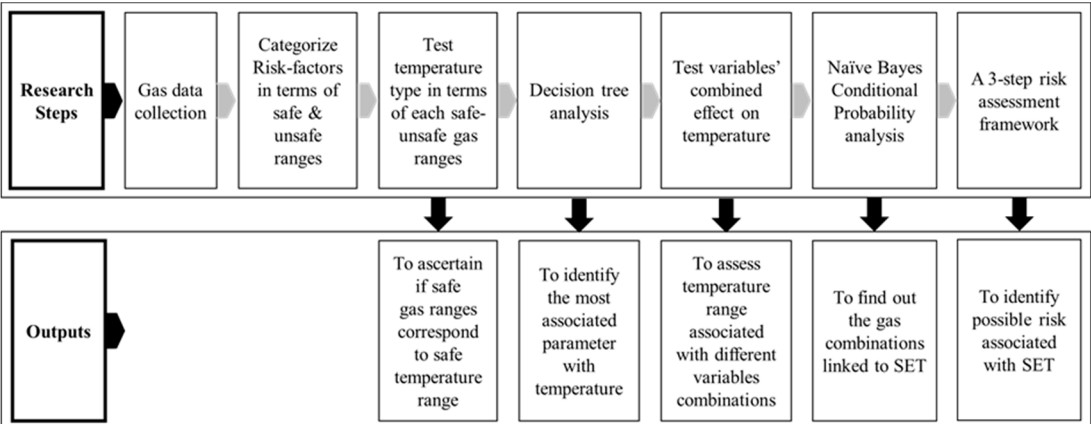

**Figure 2.** Step-by-step research procedures and outputs.

## 4. Data Collection

This research used archived data relating to the abovementioned factors collected for the Bridgeton Sanitary Landfill, Missouri. The landfill was permitted to operate in 1985 and is regulated under the Missouri Department of Natural Resources' Solid Waste Management Program (SWMP). It stopped receiving waste in 2004, when the refuse mass covered almost 210,436.5 square meters at a depth of approximately 73.2 m beneath the surface with a total 97.54 m waste thickness. The landfill first notified the SWMP about the SET events along with smoldering and odor concerns in some gas-extraction wells in 2010. Since 2013, public access was granted for commonly requested data files and reports related to subsurface smoldering events and odors in and around Bridgeton Landfill [55].

The box in the upper left-hand corner in Figure 3 shows a geographical site overview of the Bridgeton Landfill and the locations of the gas-extraction wells used to collect subsurface gas samples. The three other boxes show the names of the gas-extraction wells within the A, B and C areas outlined in the overview box. Weekly and monthly sample data are available for these gas-extraction wells and were used in this study for in-depth analysis. Weekly gas-well data comprise the concentrations of the basic gas parameters ($CO_2$, $CH_4$, $O_2$, and balance gas) and temperature data, but the monthly data contains only $CH_4$, $CO_2$, $O_2$, $N_2$, hydrogen ($H_2$), and CO without the wellhead temperature data. We used weekly available datasets for the time period of June 2013 to October 2016 since the temperature is the most important parameter for our data analysis. We collected 18,469 observations from the gas-interceptor wells, gas-collection wells, and temperature monitoring probes. Table 3 shows a sample of the collected data.

As discussed in Section 2, $RN_2$ and the $CH_4$-to-$CO_2$ ratio are two significant gas parameters to predict gas-well temperatures and can be calculated using the collected gas data. $RN_2$ is the percentage of $N_2$ during aerobic decomposition that stays unused. The overpulling of gas due to air infiltration through a gas collection system can cause there to be excess $N_2$. The $O_2$ in the intruded air pulled by the vacuum in the gas-collection system kills methanogens and initiates aerobic conditions. $O_2$ is consumed during the decomposition process and the $N_2$ present in the air remains in the landfill. A report provided by the SWANA considers $O_2$, $CH_4$, and $CO_2$ as crucial parameters to estimate the concentration of the balance gas, which primarily indicates the amount of $N_2$, and emphasizes that the usual ratio of $N_2$ to $O_2$ is approximately 3.76 [52]. Using a simple gas equilibrium equation, we estimated the $RN_2$ concentration—e.g., if a gas-well measures $CO_2$ (28.1%), $CH_4$ (32.5%), $O_2$ (3.7%), then the balance gas would be the rest of the gas portion ($100 - 32.5 - 28.1 - 3.7 = 35.7\%$). The normal $N_2$ was measured by multiplying the typical ratio (3.76) with the oxygen composition (3.7%), $3.76 * 3.7 = 13.912\%$. The $RN_2$ was then calculated by subtracting the normal $N_2$ composition (13.912%) from the balance gas (35.7%) which yields an $RN_2$ composition of 21.8% [52].

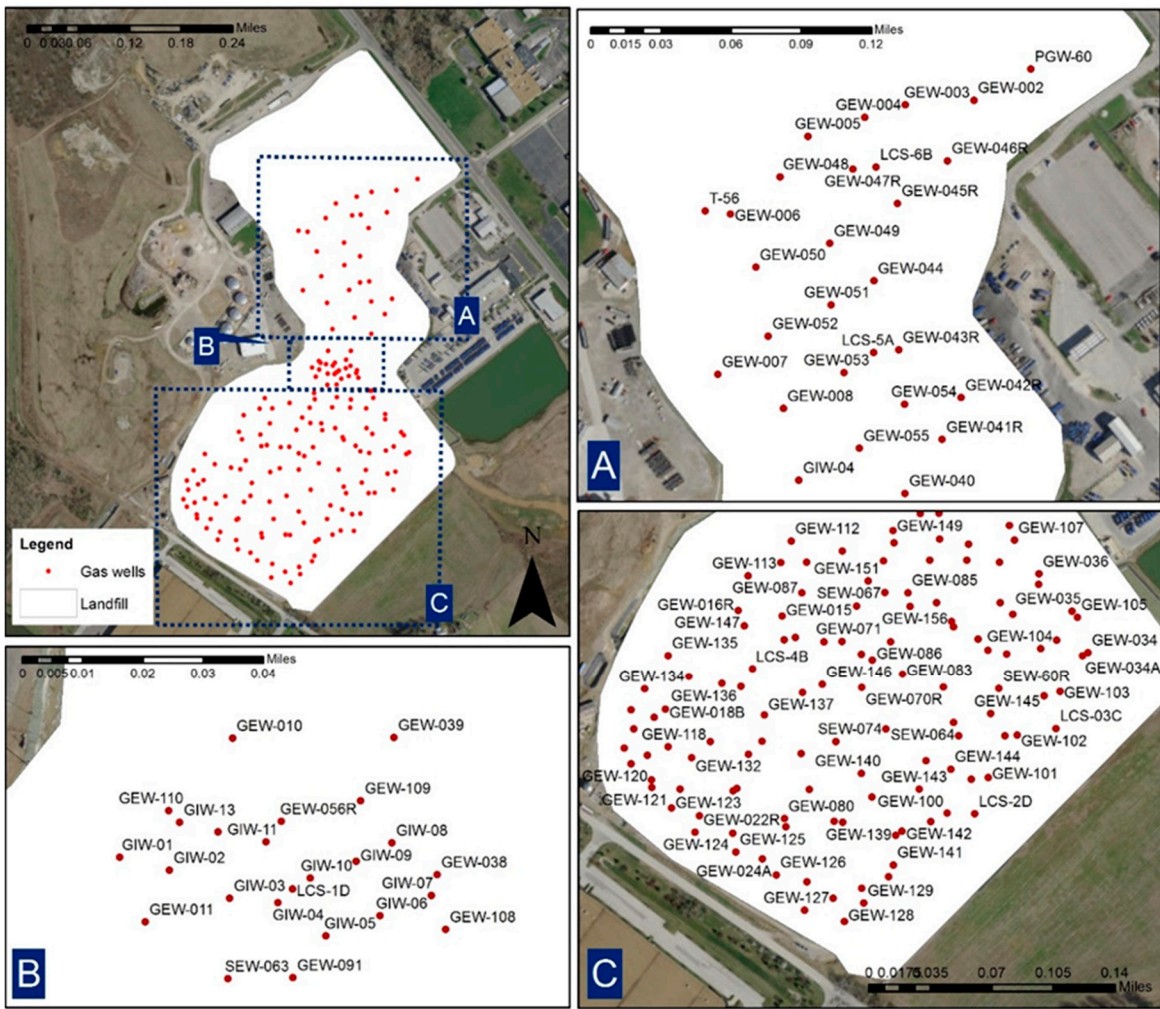

**Figure 3.** Geographical position of Bridgeton Landfill in Missouri, USA and the location of gas-extraction wells.

**Table 3.** Sample gas-well data collected for Bridgeton Landfill.

| Well Name | Date Sampled | $CH_4$ | $CO_2$ | $O_2$ | Balance Gas | Temperature (°F) | Residual $N_2$ | Ratio ($CH_2$:$CO_2$) |
|---|---|---|---|---|---|---|---|---|
| GEW-40 | 6/3/2013 9:31 | 47.9 | 51.6 | 0.0 | 0.5 | 100 | 0.5 | 0.93 |
| GEW-41R | 6/3/2013 9:35 | 57.3 | 42.2 | 0.0 | 0.5 | 116 | 0.5 | 1.36 |
| GEW-41R | 6/3/2013 9:36 | 56.8 | 41.1 | 0.0 | 2.1 | 116 | 2.1 | 1.38 |
| GEW-42R | 6/3/2013 9:39 | 53.3 | 39.9 | 0.0 | 6.8 | 112 | 6.8 | 1.34 |
| GEW-43R | 6/3/2013 9:45 | 57.4 | 42.5 | 0.0 | 0.1 | 96 | 0.1 | 1.35 |
| GEW-43R | 6/3/2013 9:46 | 56.4 | 43.5 | 0.0 | 0.1 | 140 | 0.1 | 1.30 |

## 5. Results

### 5.1. Categorization of Variables

Before we analyzed the temperature pattern in the Bridgeton Landfill, it was important to determine the correlation of the selected factors to the temperature as well as their influence on temperature. Therefore, we categorized the collected gas data in terms of the safe and normal ranges for landfill soil gases according to the operational standards mentioned in title 40 of the Code of Federal Regulations (CFR) §60.753 operational standards and considered any temperature less than 176 °F or 80 °C as a safe limit [10,39]. A SET was considered to occur when the gas wellhead temperature exceeded 80 °C, which is the highest temperature limit in any aerobic and anaerobic decomposition

process. The safe limit for $RN_2$ was considered to be less than 20% [52]. Thus, variable thresholds were established to categorize each parameter into two categories: safe and unsafe. The parameters selected for our analysis and their categorization rules are listed in Table 4.

**Table 4.** Categorization rules for the parameters based on the selected standards.

| Parameter | Rules of categorization | Category | Reference |
|---|---|---|---|
| Methane | Safe range: 45 to 60% <br> Unsafe range: < 45% and >60% | safe <br> unsafe | 40 CFR §60.753 |
| Ratio ($CH_4$:$CO_2$) | Safe range: >1 <br> Unsafe range: <1 | safe <br> unsafe | Thalhamer, 2013 |
| Oxygen | Safe range: <5% <br> Unsafe range: >5% | safe <br> unsafe | 40 CFR §60.753 |
| Temperature | Safe range: <176°F (80 °C) <br> Unsafe range: >176 °F (80 °C) | safe <br> unsafe | Thalhamer, 2013 |
| Residual Nitrogen | Safe range: <20% <br> Unsafe range: >20% | safe <br> unsafe | Estabrooks, 2013 |

For each parameter—$CH_4$, $CH_4$-to-$CO_2$ ratio, $O_2$, $RN_2$, temperature, and CO—there are two possibilities. We described each sampling event with a combination of safe or unsafe states for five factors listed in Table 4. Therefore, the total number of possible sample events in our analysis can be calculated by raising two to the number of parameters. For the five parameters in our case, the possible event numbers will be $2^5$ or 32.

*5.2. Testing the Effect of Each Variable on Temperature*

In the absence of CO data for the Bridgeton Landfill, we analyzed the other available gas parameters. First, we determined how individual gas conditions can affect the temperature. Figure 4 demonstrates the temperature ranges in boxplots, when the four factors ($CH_4$, $O_2$, $RN_2$, $CH_4$-to-$CO_2$ ratio) meet the criteria for safe and unsafe conditions (safe = 0 and unsafe = 1). The boxplot shows the distribution of temperature datapoints with five measures: the minimum, median, maximum, first and third quartile. The middle quartile marks the median value (midpoint of the data) and is indicated by the middle line dividing the box into two. The samples within the first to third quartiles contain 50% of the total samples and is represented as the interquartile range (shown as the colored boxes in Figure 4). A total of 25% of the total datapoints fall below the first quartile, while the remaining 25% fall above the third quartile. The data points considered as outliers are located above the third quartile and below the first quartile, 1.5 times outside the interquartile range and are shown as open circles in the figure. Three factors other than $O_2$ were observed to have a high median and third quartile values for temperature in the unsafe condition than in the safe condition. We noticed that the safe $O_2$ range recommended by the EPA was associated with higher temperature values typically referred to as an unsafe temperature range, which contrasts with the literature described above. To summarize Figure 4, when the gas conditions are unsafe, the temperature should be higher. Milosevic et al. [56] observed $O_2$ concentrations of 15 to 21.2% *v/v* (more than double the reference $O_2$ volume of 5%) in gas-wells, whereas the gas-well temperature ranged from 24.9 to 48.9 °C and the fire probability ratio decreased by 0.836 with an increase in the $CH_4$-to-$CO_2$ ratio concentrations. Another study [16] also found similarly high $O_2$ levels (15–21.2% *v/v*) corresponding with gas-wells in both cool temperature and SET areas. The reason for this difference in $O_2$'s impact on temperature could be the categorization of $O_2$ with reference to the recommended range by the US EPA, or its relationship to other parameters, or the operational status of this particular landfill. $O_2$'s effect on temperature should be reexamined in light of its categorization levels of varying ranges. Its reference value ($5\%_{vol}$) by the US EPA should also be reevaluated.

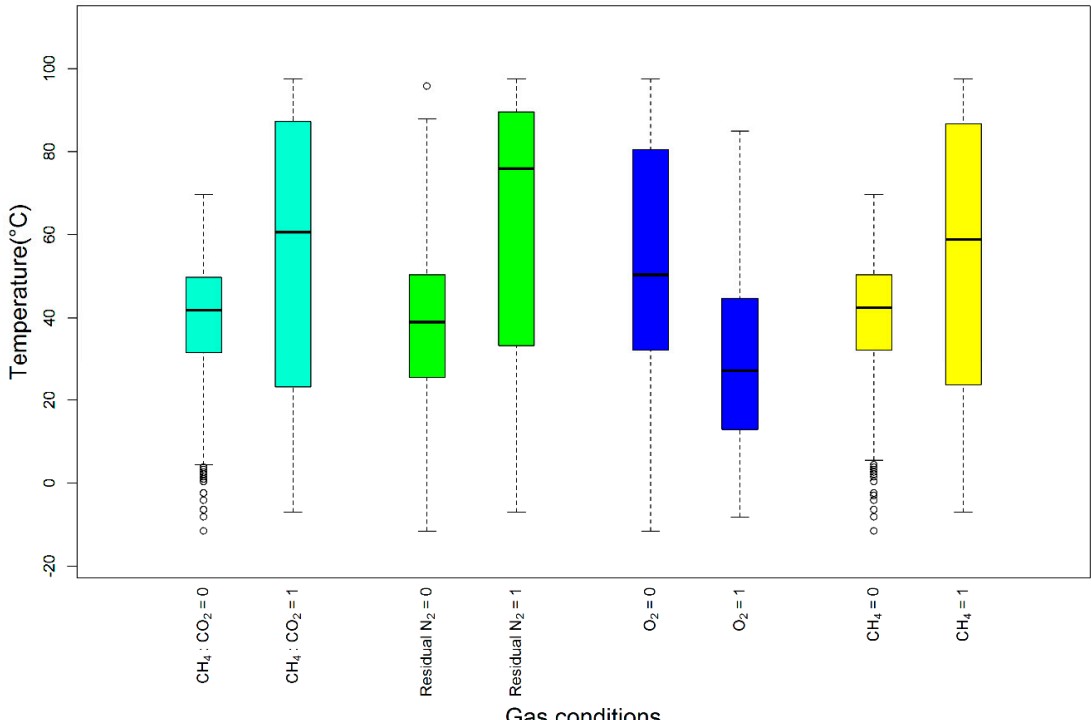

**Figure 4.** Temperature in different gas conditions.

## 5.3. Decision Tree Analysis

We applied the conditional inference trees algorithm [53] to 18,469 observation datapoints of wellhead temperatures and gas parameters to identify the factors most closely associated with subsurface temperatures. The algorithm approximates a regression relationship in a conditional inference framework using the binary recursive partitioning method. First, it tests the null hypothesis of independence between the response (temperature) and input variables (gas variables), and stops if the test fails to reject the hypothesis that temperature is an independent variable. Otherwise, the input variable with the strongest association is selected based on a *p*-value (significance level, $p < 0.05$) corresponding to the hypothesis test. Following this test, a binary partition of the selected variable is implemented. The algorithm repeats these two steps recursively at each split of a variable. The first parameter selected for the binary split represents a significant parameter generally linked to subsurface temperature.

We classified all the parameters as to whether they were in a safe or unsafe range before applying the algorithm, classifying the temperatures as "under 55 °C", "55–80 °C", "80–93 °C" or "93–149 °C". Figure 5 demonstrates all the possible splits of a decision tree at a significance level of <0.05 and the parameters with the best splits in circles with their corresponding *p*-values. The decision tree branches indicate the levels of parameters and the bar plots at the bottom show the proportions of the four temperature ranges in each end node containing all observations with a combination of features. Among the four parameters, the $CH_4$-to-$CO_2$ ratio is the covariate showing the strongest association with the temperature ranges with a significance level less than 0.001. Using a univariate logistical regression to study the synergistic effect of fire indicators, Milosevic et al. [56] found the $CH_4$-to-$CO_2$ ratio concentration to be a statistically significant fire indicator. The first tree branch starting with a ratio of <1, shows a strong association with $O_2$ (significance level, $p < 0.001$), whereas $CH_4$ was strongly associated (significance level, $p < 0.001$) with the branch of a ratio of >1. The tree shows that the branch with $CH_4$-to-$CO_2$ ratio < 1, $O_2$ < 5%, and $CH_4$ < 45% and >60% yields 8888 events from the total 18,469 data points. This branch results in the highest number of events in the temperature ranges of "80–93 °C" and "93–149 °C" than any other branches. Hence, the branch can be interpreted

as the most unsafe one. The second highest datapoints in the temperature range 80–93 °C is noticed in the branch with a $CH_4$-to-$CO_2$ ratio of <1 and $O_2$ > 5%. The third highest incidents (approximately 720 observations) in the temperature range of 55–80 °C were observed within the branch of $CH_4$-to-$CO_2$ ratio > 1 and $CH_4$ with 45–60%. Therefore, the unsafe ranges of the gas parameters are not always associated with high temperature ranges, but rather different combinations of parameters.

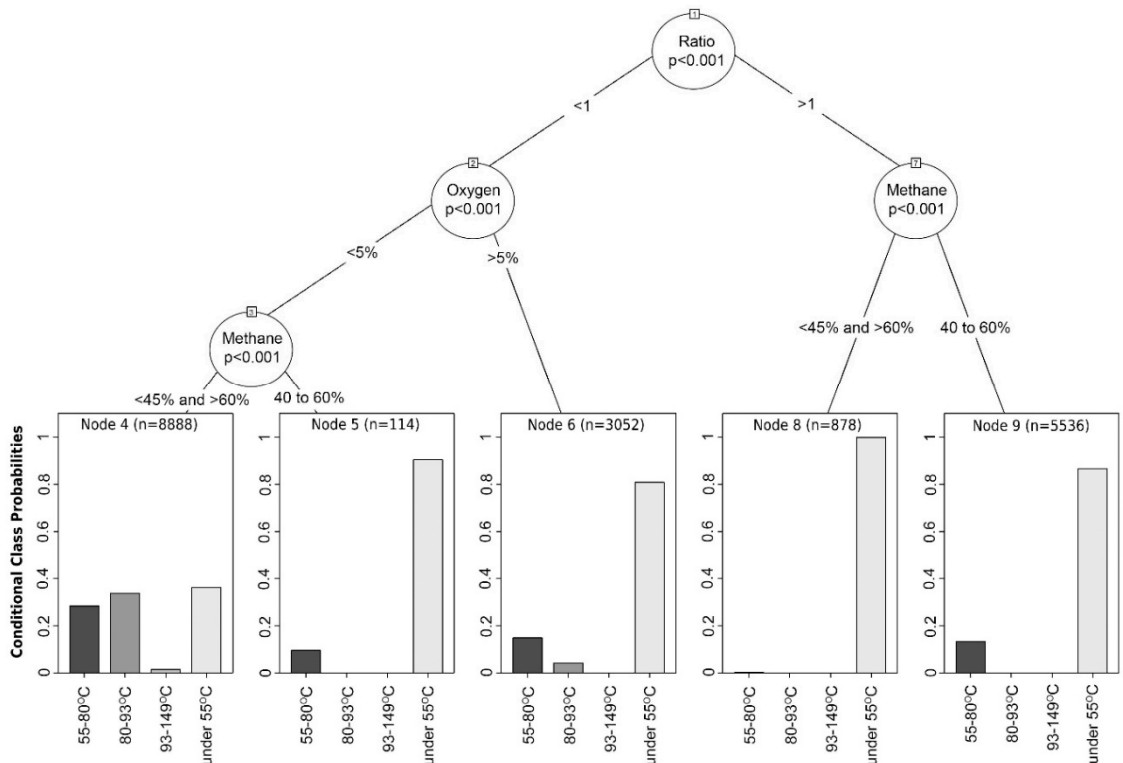

**Figure 5.** Conditional inference trees on the distribution of four temperature ranges.

### 5.4. Effect of Variable Combinations on Temperature

Although different documents or regulations regarding landfill operation suggest safe ranges for the mentioned parameters, it is essential to study how well these safe ranges corresponding with the subsurface temperature. We studied how the SET is influenced by various gas parameter combinations with a boxplot of temperature versus different events shown in Figure 6, where 0 is denoted as safe and 1 as unsafe. The observations from the figure are as follows:

- In four cases, i.e., (temperature = 0, $CH_4$ = 1, $CH_4$:$CO_2$ = 1, $O_2$ = 0, $RN_2$ = 0), (temperature = 0, $CH_4$ = 1, $CH_4$:$CO_2$ = 1, $O_2$ = 0, $RN_2$ = 1), (temperature = 0, $CH_4$ = 1, $CH_4$:$CO_2$ =1, $O_2$ = 1, $RN_2$ = 0), and (temperature = 0, $CH_4$ = 1, $CH_4$:$CO_2$ = 1, $O_2$ = 1, $RN_2$ = 1), the temperature ranges were wide. The widest interquartile temperature ranges were noticed in the events with <45% and >60% $CH_4$, ($CH_4$:$CO_2$) ratio <1, $RN_2$ <20% or >20%, $O_2$ <5% or >5% and temperature <80 °C. These combinations resulted in temperatures ranging from safe to almost unsafe, although some parameters were in the unsafe range.
- The highest positive quartile value for temperature was observed for (temperature = 1, $CH_4$ = 1, $CH_4$:$CO_2$ = 1, $O_2$ = 0, $RN_2$ = 1), which represents events with < 45% and >60% $CH_4$, ($CH_4$:$CO_2$) ratio < 1, $RN_2$ > 20%, $O_2$ < 5%, and temperature >80 °C. Here, the combination resulting in the highest SET did not include an unsafe $O_2$ range, which contradicts the suggested safe $O_2$ range.
- Only one data point was noticed for the (temperature = 0, $CH_4$ = 1, $CH_4$:$CO_2$ = 0, $O_2$ = 1, $RN_2$ = 1) combination, which indicates that the events with < 45% and >60% $CH_4$, ($CH_4$:$CO_2$) ratio > 1,

$RN_2 > 20\%$, $O_2 > 5\%$, and temperature $< 80$ °C may be rare. This combination requires more historical data to confirm its rare occurrence.

- Four of the combinations (temperature = 1, $CH_4$ = 1, $CH_4:CO_2$ = 1, $O_2$ = 0, $RN_2$ = 0), (temperature = 1, $CH_4$ = 1, $CH_4:CO_2$ = 1, $O_2$ = 0, $RN_2$ = 1), (temperature = 1, $CH_4$ = 1, $CH_4:CO_2$ = 1, $O_2$ = 1, $RN_2$ = 0), (temperature = 1, $CH_4$ = 1, $CH_4:CO_2$ = 1, $O_2$ = 1, $RN_2$ = 1) representing events with $< 45\%$ and $>60\%$ $CH_4$, ($CH_4:CO_2$) ratio $< 1$, $RN_2 < 20\%$ or $> 20\%$, $O_2 < 5\%$ or $>5\%$, and temperature $> 80$ °C showed temperatures higher than 176 °F, whereas the other combinations produced safe temperatures (under 80 °C) with a probability of more than 75%.

- Similar gas parameter combinations were noticed in three combination pairs. For example, the (temperature = 0, $CH_4$ = 1, $CH_4:CO_2$ = 1, $O_2$ = 0, $RN_2$ = 0) and (temperature = 1, $CH_4$ = 1, $CH_4:CO_2$ = 1, $O_2$ = 0, $RN_2$ = 0) combinations demonstrate same gas combinations with temperature in both safe and unsafe ranges, which indicates that temperature is insignificantly affected by gas parameters in the gas combination of ($CH_4$ = 1, $CH_4:CO_2$ = 1, $O_2$ = 0, $RN_2$ = 0). We observed a similar outcome for the gas combination of ($CH_4$ = 1, $CH_4:CO_2$ = 1, $O_2$ = 0, $RN_2$ = 1) and ($CH_4$ = 1, $CH_4:CO_2$ = 1, $O_2$ = 1, $RN_2$ = 0). There might be other missing variables that are triggering the SET with the same gas combinations. Therefore, insignificant impacts on temperature were observed in events with gas combinations such as: ($< 45\%$ and $>60\%$ $CH_4$, ($CH_4:CO_2$) ratio $< 1$, $RN_2 < 20\%$, $O_2 < 5\%$); ($< 45\%$ and $>60\%$ $CH_4$, ($CH_4:CO_2$) ratio $< 1$, $RN_2 > 20\%$, $O_2 < 5\%$); ($< 45\%$ and $>60\%$ $CH_4$, ($CH_4:CO_2$) ratio $< 1$, $RN_2 < 20\%$, $O_2 > 5\%$). In their landfill study, Jafari et al. (2017a, b) reported a trend among numerous gas-extraction wells in which a decreasing $CH_4$-to-$CO_2$ ratio occurred prior to an increase in the wellhead temperature. A decreasing $CH_4$-to-$CO_2$ ratio and a $CH_4$ level above the recommended range were noticed in these three combination pairs, which indicates that the conditions of $RN_2$ and $O_2$ do not govern the increase in temperature. Rather, other possible gases such as $H_2$ or CO may control the situation.

- Moreover, no boxplot was created for seventeen other combinations due to the absence of these combinations in the sample datapoints, which implies that these events do not occur or are extremely rare. Increasing the volume of sample data from recent years could firmly establish the truth of this statement.

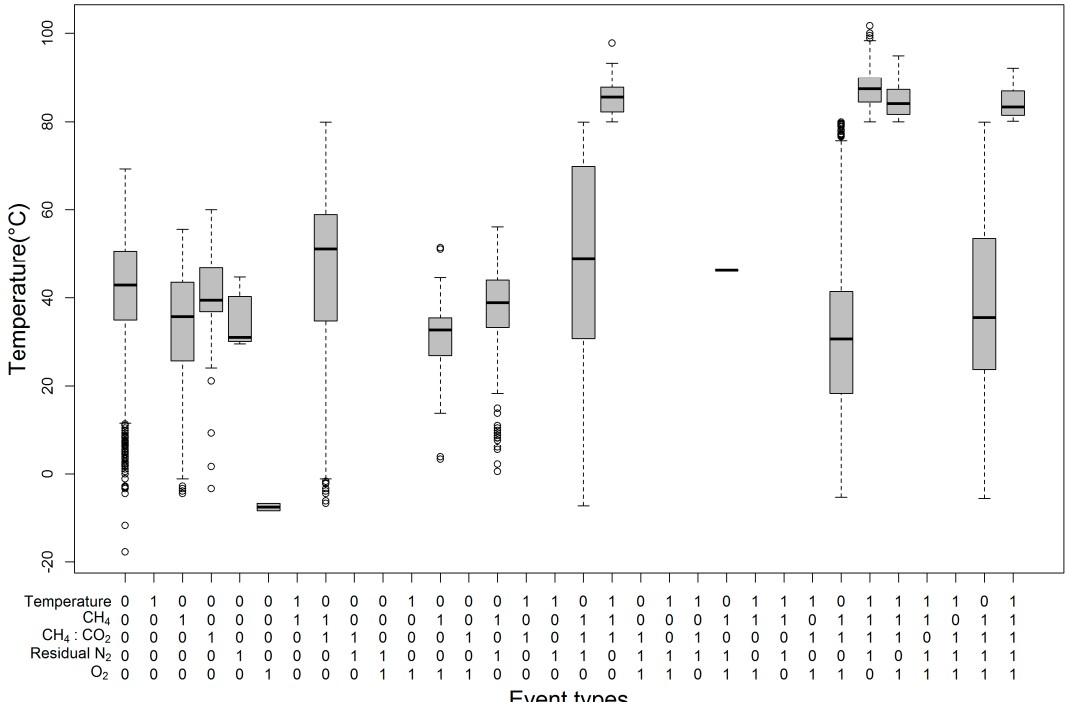

**Figure 6.** Temperature datapoints categorized in different event conditions.

We used a group histogram to plot the variance and frequency of temperature in different combinations. Figure 7 presents the temperature distribution in each of the above cases, including their spread, peaks, and symmetry. The histogram shows temperature data with different frequencies, peaks, frequently non-normal values, and outliers. Histograms include multiple peaks in some cases that indicate multimodal distributions.

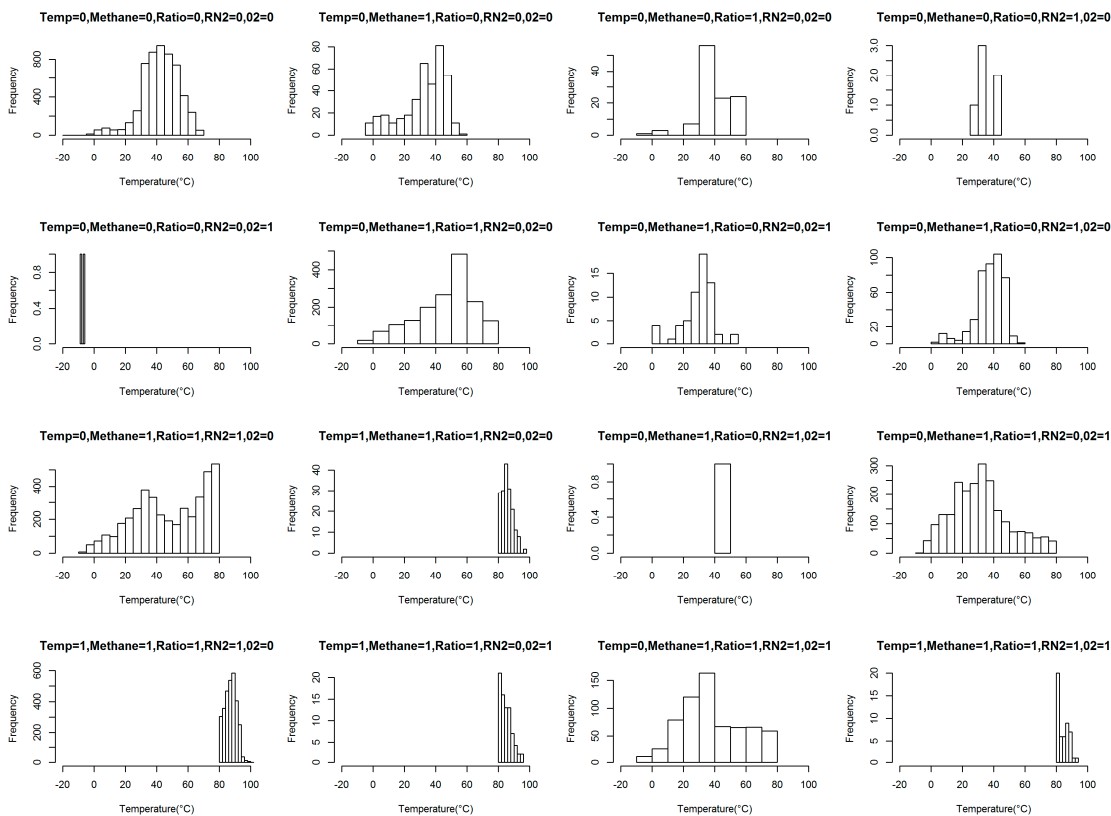

**Figure 7.** Distribution of temperature in different combinations.

The question remains whether the mean temperature range statistically varies with respect to the combinations. The Wilcoxon rank-sum test [57], a nonparametric test method, is able to interpret the interpretation of the difference between these events. The test is robust against the presence of outliers and the non-normality of a sample distribution. Tests for a null hypothesis test ($H_o$) of equal means in two independent samples were performed. All pairs of the combinations were tested with the null hypothesis of no difference between any two events. Figure 8 shows the median values of the difference between a pair of events that resulted from the Wilcoxon rank-sum tests shown in a matrix of events. Each event is represented in a sequence of the safe and unsafe condition of temperature, $CH_4$, $CH_4:CO_2$ ratio, and residual $N_2$ and $O_2$. The color ranges from green, white, and red in the figure represent a high negative difference, no difference and high positive difference between two events, respectively. For example, a pair of events (temperature = 1, $CH_4$ = 1, $CH_4:CO_2$ = 1, $RN_2$ = 1, $O_2$ = 0) and (temperature = 0, $CH_4$ = 1, $CH_4:CO_2$ = 1, $RN_2$ = 1, $O_2$ = 1) have the highest positive difference, while the (temperature = 0, $CH_4$ = 0, $CH_4:CO_2$ = 0, $RN_2$ = 0, $O_2$ = 1) events and (temperature = 1, $CH_4$ = 1, $CH_4:CO_2$ = 1, $RN_2$ = 1, $O_2$ = 0) events have the highest negative difference. In the figure, (***) represents an insignificant *p*-value <0.05 for that pair of events. The nondirectional two-sided test resulted in a significant *p*-value < 0.05 for all pairs except three pairs which indicates that the means for the three pairs of events share a significant similarity in their means and variance spreads. These three pairs are: (temperature = 1, $CH_4$ = 1, $CH_4:CO_2$ = 1, $RN_2$ = 1, $O_2$ = 0) and (temperature = 1, $CH_4$ = 1, $CH_4:CO_2$ = 1, $RN_2$ = 0, $O_2$ = 0), (temperature = 1, $CH_4$ = 1, $CH_4:CO_2$ = 1, $RN_2$ = 1, $O_2$ = 1) and

(temperature = 1, $CH_4$ = 1, $CH_4{:}CO_2$ = 1, $RN_2$ = 1, $O_2$ = 0), (temperature = 1, $CH_4$ = 1, $CH_4{:}CO_2$ = 1, $RN_2$ = 1, $O_2$ = 1) and (temperature = 1, $CH_4$ = 1, $CH_4{:}CO_2$ = 1, $RN_2$ = 0, $O_2$ = 1).

| | 00000 | 01000 | 00100 | 00010 | 00001 | 01100 | 01001 | 01010 | 01110 | 11100 | 01101 | 11110 | 11101 | 01111 | 11111 |
|---|---|---|---|---|---|---|---|---|---|---|---|---|---|---|---|
| 00000 | | 14.40 | 2.70 | 14.70 | 90.20 | -11.90 | 21.60 | 8.00 | -13.00 | -77.20 | 21.90 | -80.00 | -75.90 | 10.20 | -74.70 |
| 01000 | | | -11.00 | 2.70 | 77.99 | -27.00 | 8.60 | -5.00 | -31.60 | -89.50 | 6.00 | -92.10 | -88.00 | -6.20 | -86.60 |
| 00100 | | | | 5.00 | 17.10 | -15.00 | 17.10 | 5.00 | -15.00 | -81.00 | 19.00 | -84.00 | -79.30 | 8.20 | -78.30 |
| 00010 | | | | | 69.70 | -28.00 | 5.40 | -7.60 | -25.70 | -94.34 | 7.50 | -97.00 | -92.30 | -3.20 | -91.60 |
| 00001 | | | | | | -105.00 | -72.00 | -84.00 | -101.35 | -167.00 | -69.00 | -170.90 | -165.20 | -77.20 | -163.47 |
| 01100 | | | | | | | 35.00 | 21.00 | -4.40 | -63.00 | 31.40 | -65.80 | -61.30 | 18.50 | -60.00 |
| 01001 | | | | | | | | -13.50 | -35.90 | -97.00 | 0.00 | -100.00 | -95.70 | -10.20 | -94.70 |
| 01010 | | | | | | | | | -22.00 | -84.00 | 14.00 | -87.00 | -83.00 | 3.10 | -81.90 |
| 01110 | | | | | | | | | | -66.00 | 31.10 | -68.80 | -64.70 | 17.60 | -63.40 |
| 11100 | | | | | | | | | | | 99.30 | -3.00 | 1.50 | 90.00 | 2.50 |
| 01101 | | | | | | | | | | | | -102.20 | -98.00 | -11.40 | -96.90 |
| 11110 | | | | | | | | | | | | | 4.80 | 92.90 | 0.80 |
| 11101 | | | | | | | | | | *** | | | | 88.50 | 0.80 |
| 01111 | | | | | | | | | | | | | | | -87.60 |
| 11111 | | | | | | | | | | | | *** | *** | | |

**Figure 8.** Median values of the difference between a pair of events resulted from the Wilcoxon rank-sum tests shown in a matrix of events (note: *** represents an insignificant *p*-value < 0.05 for that pair of events).

Furthermore, the probability of four temperature ranges (under 55 °C, 55–80 °C, 80–93 °C, 93–149 °C) were analyzed with respect to the different combinations of four gas parameters in the series of $CH_4$, $CH_4$-to-$CO_2$ ratio, $RN_2$, and $O_2$. The results of this analysis help us to identify possible gas combinations linked to SETs. Figure 9 shows a gradual upward trend for the 80–93 °C range in the four gas combinations of 1_1_0_1, 1_1_0_0, 1_1_1_0, and 1_1_1_1; whereas a decreasing trend is observed for the "under 55 °C" range. The 1_1_1_0 combination has a likelihood of only 3% in the 93–149 °C range. Hence, the graph shows that the 1_1_1_0 combination has the greatest potential to associate with high temperature ranges, rather than the 1_1_1_1 combination in which all the gas parameters fall in the unsafe range. Temperatures in the 55–80 °C range occur with a 15% probability in the combinations with 1_1_1_0, 1_1_1_1, 1_1_0_0, 1_1_0_1, 1_0_1_0, 1_0_1_1. Therefore, gas combinations with 1_1_0_0, 1_1_0_1, 1_1_1_0, 1_1_1_1 should be considered high risk; 1_0_1_0, 1_0_1_1 combinations correspond to a medium risk that indicates a tendency to proceed to a high risk level. Table 5 summarizes the potential risk levels (low, moderate, or high) corresponding with different gas combinations based on the 80 °C safe limit discussed in Section 5.1, which could be informative for the assessment of an SET risk.

**Table 5.** Potential risk levels for different gas combinations.

| Gas Combinations ($CH_4$_$CH_4$-to-$CO_2$ Ratio_$RN_2$_$O_2$) | Risk Levels |
|---|---|
| 0_0_0_0, 0_1_0_0, 1_0_0_0, 1_0_0_1, 0_0_0_1, 0_0_1_0 | Low |
| 1_0_1_0, 1_0_1_1 | Moderate |
| 1_1_0_0, 1_1_0_1, 1_1_1_0, 1_1_1_1 | High |

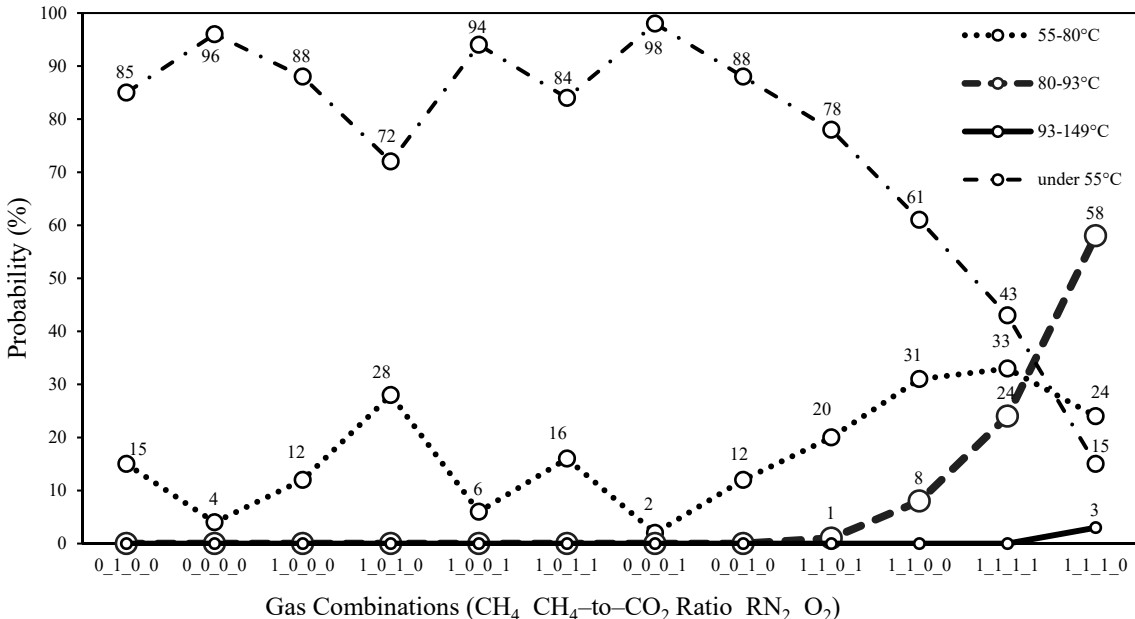

**Figure 9.** Naïve Bayes conditional probability [54] of four temperature ranges corresponding with diffent gas combinations.

## 6. Discussions

Our main goal of this study was to investigate the effects of soil gases on the SET over a certain threshold. There are several regulations recommended by the regulatory agencies regarding acceptable ranges of these landfill soil gases and subsurface temperatures. A temperature threshold of 80 °C was selected in this study since temperatures up to 80 °C have been observed during normal biological decomposition processes. We selected the thresholds for some of the gas parameters according to those reported in the 40 CFR §60.753 operational standards—Thalhamer [39] and Estabrooks [52]. Our observations from the statistical analysis conducted above, can be drawn:

- Unsafe temperature events have occurred more in the unsafe range for $CH_4$, the $CH_4$-to-$CO_2$ ratio, and $RN_2$, but not for $O_2$. The combination associated with the highest SET did not include an unsafe $O_2$ range, a finding which contradicts the suggested safe $O_2$ range.

- The conditional inference tree algorithm shows that the $CH_4$-to-$CO_2$ ratio of the four parameters has the strongest association with temperature. Ratios less than 1, $O_2$ concentrations less than 5% and $CH_4$ <45% and >60% resulted in the highest probability of events ranging from 80 to 149 °C. However, the safe $O_2$ range was not associated with a safe temperature range, which is also consistent with the observation above.

- Some combinations of gas parameters, i.e., ($CH_4 = 1$, $CH_4$:$CO_2 = 1$, $O_2 = 0$, $RN_2 = 0$), ($CH_4 = 1$, $CH_4$:$CO_2 = 1$, $O_2 = 0$, $RN_2 = 1$), and ($CH_4 = 1$, $CH_4$:$CO_2 = 1$, $O_2 = 1$, $RN_2 = 0$) have an insignificant effect on temperature. This implies that there are other confounding variables affecting these three combinations.

- Only the ($CH_4 = 1$, $CH_4$:$CO_2 = 1$, $O_2 = 0$, $RN_2 = 1$) combination shows a high occurrence in the 93–149 °C range based on the naïve Bayes conditional probability, whereas a gradual upward trend is observed for the 80–93 °C range in the four combinations of ($CH_4 = 1$, $CH_4$:$CO_2 = 1$, $O_2 = 1$, $RN_2 = 0$), ($CH_4 = 1$, $CH_4$:$CO_2 = 1$, $O_2 = 0$, $RN_2 = 0$), ($CH_4 = 1$, $CH_4$:$CO_2 = 1$, $O_2 = 1$, $RN_2 = 1$), and ($CH_4 = 1$, $CH_4$:$CO_2 = 1$, $O_2 = 0$, $RN_2 = 1$). Using the conditional probability results, the gas combinations have been summarized according to their potential risk levels (low, moderate, or high) with respect to SETs.

- The unsafe ranges of the gas parameters are not always associated with high temperature ranges, but rather different combinations of parameters.

- A three-step procedure can be followed to evaluate the risk of a landfill subsurface fire. The diagram in Figure 10 provides a general idea of this step-by-step procedure for preventing subsurface fire incidents. Step 1 is to check the temperature to determine if it falls within the unsafe range, which would require that other parameters be controlled to bring the temperatures into the safe range. Otherwise, landfill personnel should proceed to step 2 which involves checking the gas combinations. Other preventive measures should be considered if one of the four combinations of ($CH_4 = 1$, $CH_4:CO_2 = 1$, $O_2 = 1$, $RN_2 = 0$), ($CH_4 = 1$, $CH_4:CO_2 = 1$, $O_2 = 0$, $RN_2 = 0$), ($CH_4 = 1$, $CH_4:CO_2 = 1$, $O_2 = 1$, $RN_2 = 1$) and ($CH_4 = 1$, $CH_4:CO_2 = 1$, $O_2 = 0$, $RN_2 = 1$) occurs. Step 3 proposes to check gas-wells showing similar combinations of ($CH_4 = 1$, $CH_4:CO_2 = 0$, $O_2 = 0$, $RN_2 = 1$) and ($CH_4 = 1$, $CH_4:CO_2 = 0$, $O_2 = 1$, $RN_2 = 1$) that correspond to a medium risk, and then monitor these gas-well locations more carefully or more often to ensure that they do not become one of the riskiest combinations. Landfill authorities can repeat this three-step action plan as part of their regular monitoring practices of the gas collection system.

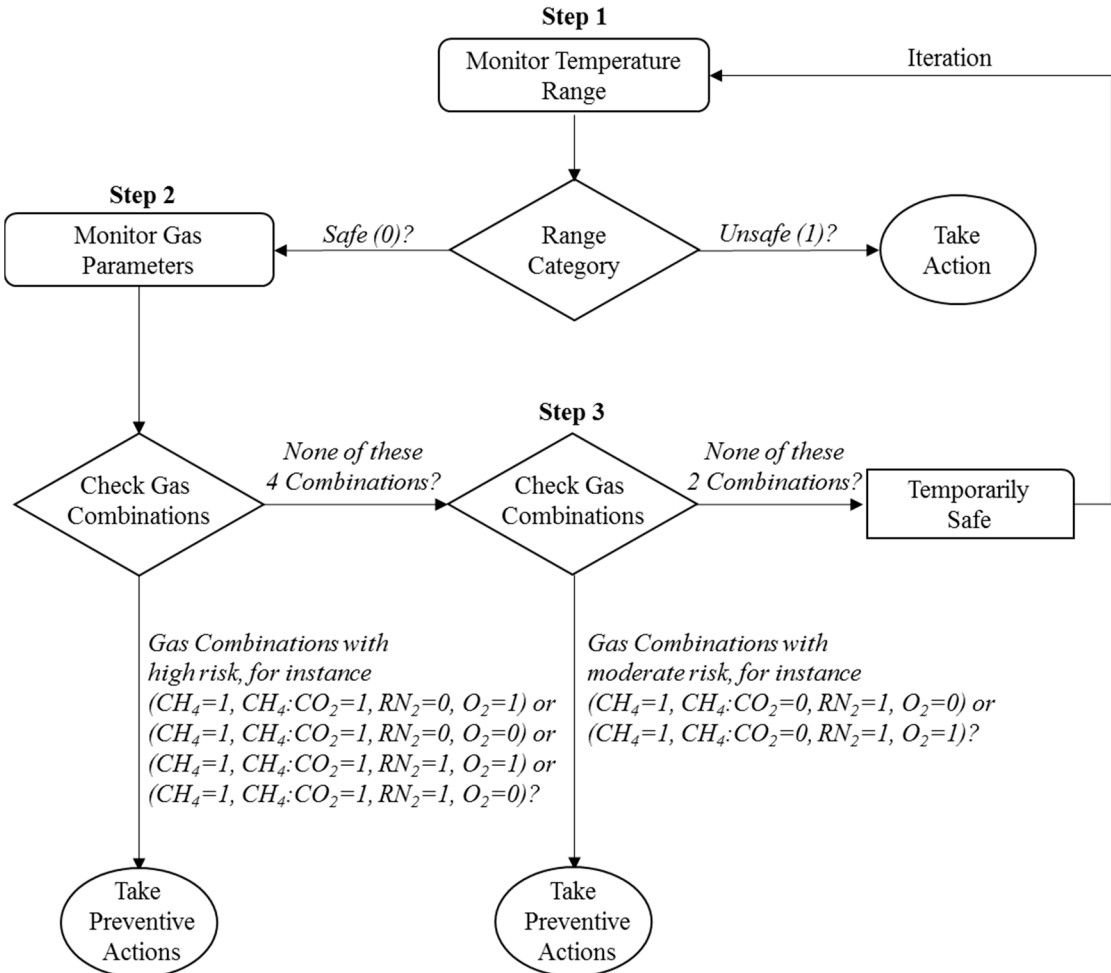

**Figure 10.** A three-step action plan for preventing landfill subsurface fires.

The dataset used for this research does not contain CO data, which may be a significant parameter affecting landfill subsurface fire incidents. Jafari et al. [16] reported observations of CO concentrations above 1000 ppmv and up to 10200 ppmv at wellhead temperatures above 68 °C and as high as 95 °C. The reviewed literature also acknowledges CO as a significant indicator for identifying SETs. The analysis of gas variables without the incorporation of CO data might overlook scenarios that could be linked to SETs. Hence, adding CO to the gas combinations will lead to a more precise identification of high-risk scenarios. Similarly, incorporating parameters such as $H_2$ concentrations, leachate collection, and pressure could improve the analysis results.

## 7. Conclusions

The objective of this article relates to identifying the causes of landfill fire hazards associated with the major landfill gases. Currently, the system lacks appropriate and standardized measures in the forecasting and controlling of subsurface landfill fires and lets the individual landfill authority or owner deal with this hazard on their own. This research developed a methodological approach to this issue and attempted to evaluate the recommended safe ranges of the landfill gases and gas-well temperatures. The relationships were investigated between the subsurface elevated temperature (SET) and soil gases, with the greatest influence from landfill gases (i.e., methane, carbon dioxide, carbon monoxide, nitrogen, and oxygen). Our works observed that SET events do not usually occur in the unsafe oxygen range; similarly, the safe oxygen range does not always associate with the safe temperature range. Rather than focusing on the individual gas effects, we observed temperature behaving differently with the different combinations of gases in safe or unsafe ranges. Our research identified some gas combinations highly associating with SET events and proposed a three-step procedure to control the SET events before they progress into subsurface fires. The research methodology used in this study can be repeated with respect to the parameter thresholds established by different regulatory agencies, i.e., the United States Environmental Protection Agency (US EPA), Solid Waste Association of North America (SWANA), International Solid Waste Association (ISWA), and United States Army Corps of Engineers (USACE). The effect of unsafe temperature conditions can be examined by comparing the results obtained with respect to these regulatory agencies. However, our proposed methodology might be a useful tool for the landfill authorities in regulating subsurface landfill temperatures.

**Author Contributions:** Conceptualization, S.S. and R.N.; Methodology, S.S., J.W.E. and R.N.; Software, S.S. and M.G.R.F.; Validation, S.S. and M.G.R.F.; Formal Analysis, S.S.; Investigation, S.S.; Resources, S.S.; Data Curation, S.S. and M.G.R.F.; Writing—Original Draft Preparation, S.S.; Writing—Review & Editing, S.S., J.W.E., R.W.P., R.N. and M.K.; Visualization, S.S.; Supervision, J.W.E. and R.N.; Proofreading, R.W.P., R.N. and M.K.; Project Administration, R.N.; Funding Acquisition, R.N. All authors have read and agreed to the published version of the manuscript.

**Funding:** This work was funded by the US Department of Agriculture (USDA) Solid Waste Management Grant Programs.

**Conflicts of Interest:** The authors declare no conflict of interest.

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
