# Peer review of "Investigating Effects of Landfill Soil Gases on Landfill Elevated Subsurface Temperature"

_applsci, doi:10.3390/app10186401_

Round 1

Reviewer 1 Report

The topic of the manuscript “Investigating Effects of Landfill Soil Gases on Landfill Elevated Subsurface Temperature” is interesting and the authors had a good and innovative idea for a research project. The subject is relevant, the analytical methodologies are adequate, and the volume of data seems to be enough for publication. Methodology is well explained. I have no hesitation in recommending publication following minor revision.

Abstract: Abstract really presents summary, include key findings and the length of this part of the manuscript is appropriate.

Introduction: I consider that the structure of this section was well designed. Literature Review is adequate. Is effective, clear and well organized however, authors should focus on new literature (not older than 5 years).

What about hypothesis of the research? What is the research question?

Material and methods: The methodology is well thought through.

Results and discussion: The results of the study are well presented.  However, should be more conclusive and also generalised confirming if they have wide validity.

Conclusion: In my opinion, this part of the work is too extensive. It is also not customary to refer to literature in conclusions.

The aim, range and results were clearly defined. The work contains appropriate testing methods and analyses of results. In my opinion authors should stress and demonstrate sufficient scientific novelty of this research.

Reviewer 2 Report

The objective of this research is to develop a methodological approach to identify the causes of landfill fire hazards. A Naïve Bayes technique for conditional probability was implemented to investigate how different gas combinations can affect different temperature ranges with respect to the safe and unsafe states of these gases. The results indicate that methane and carbon dioxide gases are strongly associated with subsurface elevated temperature. There are some comments and suggestions for authors.

  1. In the Fig. 5, there are many statistical results. What is the hypothesis the authors used in this manuscript? What kind of the method are used to test the statistical results?
  2. In the decision tree, how to generate the results ( ex. The tree shows that the branch with CH4-to-CO2 ratio < 1, O2 <5%, and CH4 <45% and >60% has the highest number of datapoints in the temperature ranges of ‘55–80 °C,’ ‘80–93 °C,’ and ‘93–149 °C’ than any other branches)? Please describe the method clearly.
  3. Please shows the results of Wilcoxon rank-sum test and illustrates the hypothesis.
  4. In the conclusion section, the authors focus on too many results. I suggest the authors to shorten the conclusion to emphasize the contribution of this manuscript.

Round 2

Reviewer 2 Report

Accept in present form